

# Molecular characterization of *Pectobacterium atrosepticum* infecting potato and its management through chemicals

Akhtar Hameed[1], Muhammad Zeeshan[1], Rana Binyamin[1],
Muhammad Waqar Alam[2], Subhan Ali[1], Muhammad Saqlain Zaheer[3],
Habib Ali[3], Muhammad Waheed Riaz[4], Hafiz Haider Ali[5,6],
Mohamed Soliman Elshikh[7] and Khaloud Mohammed Alarjani[7]

[1] Institute of Plant Protection, MNS-University of Agriculture Multan, Multan, Punjab, Pakistan
[2] Department of Plant Pathology, University of Okara, Okara, Pakistan
[3] Department of Agricultural Engineering, Khwaja Fareed University of Engineering and Information Technology, Rahim Yar Khan, Pakistan
[4] State Key Laboratory of Wheat Breeding, Group of Wheat Quality and Molecular Breeding, College of Agronomy, Shandong Agricultural University, Tai'an, Shandong, China
[5] Crop, Soil, and Environmental Sciences, University of Arkansas, Fayetteville, AR, USA
[6] Department of Agriculture, Government College University Lahore, Lahore, Pakistan
[7] Department of Botany and Microbiology, College of Science, King Saud University, Riyadh, Saudi Arabia

Corresponding authors
Akhtar Hameed, akhtar.hameed@mnsuam.edu.pk
Muhammad Saqlain Zaheer, msaqlainzaheer@gmail.com

## ABSTRACT

Potato farming is a vital component of food security and the economic stability especially in the under developing countries but it faces many challenges in production, blackleg disease caused by *Pectobacterium atrosepticum* (*Pa*) is one of the main reason for damaging crop yield of the potato. Effective management strategies are essential to control these losses and to get sustainable potato crop yield. This study was focused on characterizing the *Pa* and the investigating new chemical options for its management. The research was involved a systematic survey across the three district of Punjab, Pakistan (Khanewal, Okara, and Multan) to collect samples exhibiting the black leg symptoms. These samples were analyzed in the laboratory where gram-negative bacteria were isolated and identified through biochemical and pathogenicity tests for *Pa*. DNA sequencing further confirmed these isolates of *Pa* strains. Six different chemicals were tested to control blackleg problem in both *vitro* and *vivo* at different concentrations. *In vitro* experiment, Cordate demonstrated the highest efficacy with a maximum inhibition zones of 17.139 mm, followed by Air One (13.778 mm), Profiler (10.167 mm), Blue Copper (7.7778 mm), Spot Fix (7.6689 mm), and Strider (7.0667 mm). *In vivo*, Cordate maintained its effectiveness with the lowest disease incidence of 14.76%, followed by Blue Copper (17.49%), Air One (16.98%), Spot Fix (20.67%), Profiler (21.45%), Strider (24.99%), and the control group (43.00%). The results highlight Cordate's potential as a most effective chemical against *Pa*, offering promising role for managing blackleg disease in potato and to improve overall productivity.

## INTRODUCTION

Potato (*Solanum Tuberosum* L.) belongs to the family Solanaceae, and a rich source of starch and protein. In terms of human consumption, potato is at third number after rice and wheat (*Rahaman & Shehab, 2019*). It is the world's first non-cereal crop, producing 359.1 million tons annually over 16.5 million per ha, with an average yield of 21.77 t per ha in 2020–21 (*FAO, 2022*). In Pakistan, potato area, production and average yield have been 174.4 thousand hectares, 4,245.9 thousand tones, and 24.3 t/ha, respectively during 2017–18 (*Ijaz-ul-Hassan, Khan & Hussain, 2022*). The potato is used as a staple food in many countries around the world, feeding more than 10 million people (*Majeed & Muhammad, 2018*). All essential elements, including carbohydrates, amino acids, several vitamins (such as vitamin C and vitamin A), numerous minerals including (potassium, magnesium, and iron), beneficial phytochemicals (such as phenolics, flavonoids, and carotenoids), and dietary fiber (skin) can be full filed by potato tubers. Therefore, it is known as a complete food because it has all of the crucial dietary components needed for growth and health (*Hussain, 2016*). The reasons behind the spread of potatoes all over the world are their rich nutrient contents, high yield potential, and high adaptability. It is recommended as a food security crop because many countries are facing the big problem of rising population and food supply issues (*Devaux, Kromann & Ortiz, 2014*, *Devaux et al., 2021*). Hence, potato-based farming practices provide increasingly important potential for improving health, poverty reduction, and food security for rural people (*Devaux, Kromann & Ortiz, 2014*, *Devaux et al., 2021*). Two-third of the fresh potatoes are roughly consumed after harvest, and about 1.3 billion people consume potatoes yearly as a staple food (more than 50 kg per person) including India and China (*Devaux et al., 2021*). The world average for annual potato consumption per person was computed at 35 kg in 2013, with Europe showing a progressive decline but developing countries scaling up by nearly more than 50% (29 kg in Asia and 19 kg in Africa) (*Wijesinha-Bettoni & Mouillé, 2019*).

According to trends in the industry, the production of potatoes has been shifted from countries that are developed toward less developed or developing countries. In 2005, developing countries produced more potatoes as compared to developed nations (52% of total production) (*FAOSTAT, 2018*).

Potato farming plays a crucial role in addressing food security and poverty reduction. It is a vital food and cash crop, with significant potential to contribute to global food security, particularly in developing countries. However, the sustainability of potato production is threatened by various challenges, including adverse abiotic conditions, pests, and pathogens (*Waals & Krüger, 2020*).

*Pectobacterium atrocepticum* is a quarantine and threatening phytopathogen causing blackleg disease of potatoes in many countries (*Waals & Krüger, 2020*; *Rivedal, Brazil & Frost, 2021*). By affecting potato yield, tuber quality as well as food biosecurity, *P. atrosepticum* is one of the most dangerous potato pathogens and can lead to significant

economic losses. A higher percentage of vulnerable types have been planted, and climatic change has altered the genetic composition of the population of pathogens that causes blackleg and soft rot in southern and to lessen its impact in Zimbabwe. These factors are likely the causes of the disease's increasing severity and dissemination (*Waals & Krüger, 2020*).

The blackleg disease is more prevalent under wet conditions with soil and environmental temperature of 10–18 °C minimum and 15–26 °C maximum. The pathogens spread takes place in many ways with infected mother tuber playing an important role. The bacterium has been observed to survive on tuber skin, in lenticels, and in small wounds which may have been introduced during postharvest handling (*Czajkowski et al., 2011*; *Dupuis, Nkuriyingoma & Van Gijsegem, 2021*; *Been, Beniers & van der Wolf, 2022*).

*Pa* cells can also arise from other plant hosts or be transmitted by different vectors, such as insects, maggots, or nematodes (*Rossmann et al., 2018*; *Toth et al., 2021*). Additional sources for this disease include waterways (*McCarter-Zorner et al., 1984*), soil (*Peltzer & Sivasithamparam, 1988*), air, aerosols, and surfaces of field machinery or storage containers (*Toth et al., 2021*). The pathogens spread takes place in many ways with infected mother tuber playing an important role. The bacterium has been observed to survive on tuber skin, in lenticels, and in small wounds which may have been introduced during postharvest handling (*Been, Beniers & van der Wolf, 2022*). It was identified for the first time in the Swat Valley of Pakistan in 1984 and caused crop failure which led to a crop loss by 45% (*Rashid et al., 2012*).

Blackleg of potato tubers has become one of the most widespread diseases on the planet (*Sarfraz et al., 2018*). *Javed (1995)* identified blackleg disease in different zones of Pakistan. It is reported that disease develops in potato plants when the moisture content of plants becomes high. *Riaz et al. (2008)* reported that 40% of disease occurs in the summer and spring seasons. The survey was performed in KPK by collecting 42 isolates and the frequency of relevant bacterium was found accordingly: *Pectobacterium chrysanthemi* 7%, *Pectobacterium atrosepticum* 48% and *Pectobacterium carotovorum* 45%.

The blackleg disease in potatoes is the major issue that should always be addressed because it has the most adverse effects on the crop quality and yield. Blackleg may cause the low grading or even refusal of seed crystal, which in turn can cause monetary losses for the farmers. Additionally, the bacteria causing blackleg can also cause soft rotting in tubers, further reducing the market value of the crop (*Waals & Krüger, 2020*).

DNA-based methods are employed to identify and understand bacteria. Among these, the 16s rDNA universal primers are widely used for quick diagnoses. Many strains of *Pectobacterium atrosepticum* have been analyzed genetically, revealing that they are closely related to each other and to the species *Pectobacterium atrosepticum* (*Avrova et al., 2002*). However, pointed out that despite this close similarity, the genetic factors that explain why different members of *Pectobacterium* spp. infect distinct vegetable crops have not been fully understood yet. Controlling blackleg of potato is a real challenge for producers. Many control strategies have been developed and applied. Extensive research has been conducted on management techniques for *Pa*-induced disease. However, the availability of effective

commercial control treatments for soft rot and blackleg diseases remains highly limited (*Van der Wolf et al., 2021*). In agriculture, the conventional approach that primarily emphasizes seed certification is commonly utilized. This involves employing various physical and chemical treatments, such as hot water, hot dry air, steam, UV, or antibiotics, to diminish the population of Pectobacterium in potatoes that are latently infected (*Charkowski, 2015*).

This present study aimed to isolate blackleg associate pathogen found in Punjab Province of Pakistan and to identify these isolates through molecular characterization, as well as to test some new chemistry fungicides to control the disease.

## MATERIALS AND METHODS

### Survey

The survey was conducted in three major potato producing districts of Punjab Province including Okara (30.719990 N, 73.811831 E), (30.739494 N, 73.835778 E), Khanewal (30.20072 N, 72.14366 E), (30.321688 N, 72.264657 E), (30.353590 N, 72.270321 E), (30.489499 N, 72.265164 E), (30.485484 N, 72.294355 E) and Multan (30.160340 N, 71.448581 E), (30.142425 N, 71.451434 E), (30.119715 N, 71.442275 E). Each area was surveyed at intervals of 5–10 km along roadsides, with five places and five samples taken in each field using a random and zigzag approach of Autumn season crop in January 2023. The Okara district was previously very well known for potato growing but nowadays Khanewal and Multan are also counting for potatoes because of changes in environmental conditions. Plants that are showing characteristic symptoms of blackleg disease including black collar of stem and rotting around lower stem which leads to decay of plant as described by *Hashemi Tameh et al. (2020)* were collected. Collected samples were labeled with the date, Location, and time of sample collection and were brought into the Diagnostic lab of MNS-University of Agriculture, Multan.

### Isolation, purification and multiplication

Stem showing typical characteristics of blackleg symptoms were collected in brown paper bags (10″ x 12″) and then taken to the Diagnostic Lab for isolation of *Pa* by using the streaking method (*Rehman et al., 2015*). Firstly, a sterilized beaker with a capacity of 1 liter was taken and 14 g of nutrient agar (CM0003; OXOID) and 500 ml of distilled water were added in it and autoclaved the media at 121 °C and 15 psi for 15 min using autoclave (JSAX-60) (*Riaz et al., 2008*). The media were then carefully poured into sterilized petri plates in a laminar flow chamber (RTVL-1312; Robus, London, United Kingdom) and placed for 15 min for solidification. Infected stem portion was carefully cut of 1 cm in length (approx.) with a small healthy portion and surface sterilized in 1% sodium hypochlorite for 1 min followed by three sequential rinses with sterile distilled water (*Palafox-Leal et al., 2023*). After surface sterilization, moisture was removed by putting the samples on blotter paper. These samples were placed onto NA media plates and wrapped the plates with wrapping tape. After proper tagging, the plates were kept in the Bacteriological Incubator (Memmert Type; Servo) for 24–36 h at 28 °C to encourage bacterial growth. After 24–36 h, bacterial colonies around the samples on the NA media

plates were observed and streaking technique was used to transfer them onto new sterilized plates with NA medium. After 24 to 36 h of incubation, round colonies with a yellow hue emerged. A pure single culture of bacterium was stored at $-80\ °C$ in a refrigerator (ZLN-T 300 COMF) with 50% glycerol.

## Identification of pathogens

The bacterium that grew on the culture plate was identified morphologically by observing its color and type of colony. The biochemical tests such as Gram staining and 3% KOH test were conducted as described by *Holt et al. (2000)* and *Mubeen et al. (2015)*.

## Pathogenicity tests

For pathogenicity, the bacterial culture of all isolates was grown overnight in 45 mL test tubes containing nutrient broth (NB) at 28 °C by following *Hoque & Mansfield (2005)* protocol and placed on a shaking incubator (NB-205LF) at 250 rpm. By using spectrometer (96 micro well plate reader BioTek, model; H-QuantTM, Winooski, VT, USA), the resulting bacterial suspension was adjusted at $10^8$ cfu/mL. The pathogenicity was performed with Koch's postulates and the Tuber assay test.

## Koch's postulates

Confirmation of all the isolated *Pa* isolates was done by fulfilling Koch's postulates (*Juhasz et al., 2013*). Potato plants were planted in pots using the Completely Randomized Design (CRD) method in a control condition at temperature ranging from 26–29 °C. After 30 days of sprouting, the bacterial suspension (about 2 µl) was inoculated in the potato plants with three replications using the pin-pick method (*Francis, Peña & Graham, 2010*). After 15 days of inoculation, data were recorded and re-isolation of the bacterium from diseased leaves was performed to fulfill the Koch's postulate. The entire experiment was repeated twice and managed in a greenhouse and plants were kept in the greenhouse for 2 weeks (*Amaral et al., 2010*).

## Potato tuber assay

Potato tubers were surface sterilized with 3% sodium hypochlorite (NaOCl) solution, afterward about 3 mm deep wound was created on the tubers followed by the addition of 10 µl bacterial suspension ($10^8$ cfu/ml) by following *Palafox-Leal et al. (2023)*, and the wound was closed with the help of yellow tips (*Palafox-Leal et al., 2023*). Tubers were merely injected with pure NA broth as a control. After being kept in an incubator at 28 °C for three to 4 days, treated tubers were examined for symptoms of deterioration.

## DNA extraction and PCR

The DNA of the bacterial isolate was extracted by using the modified CTAB method (*Kalia, Rattan & Chopra, 1999*) and DNA pellets were washed with ethanol and dried for PCR. DNA amplification was performed with Pcc3F/Pcc3R (*Kabir, Taheri & Dumenyo, 2020*), followed by screening with ExpccF/ExpccR (*Kang, Kwon & Go, 2003*) and BR1f/L1r (*Duarte et al., 2004*). In PCR protocol denaturation (94 °C), annealing (50 °C), and extension (68 °C) were adjusted for 1 min, 1 min, and 2 min for 34 cycles, respectively, and

**Table 1  Currently used chemicals against *Pectobacterium atrosepticum*.**

| No. of Products | Market name | Active ingredients | Brand/Company |
|---|---|---|---|
| 1. | Air One | Copper oxychloride + Copper hydroxide 20%SC | Swat agro chemicals |
| 2. | Blue Copper | Copper oxychloride | Syngenta |
| 3. | Profiler | Flupicolide + fosetyle-Al | Buyer crop sciences |
| 4. | Cordate | Kasogamycin | Kanzo Ag |
| 5. | Strider | Validamicin | Agrarian crop sciences |
| 6. | Spot Fix | Oxine copper | Tara group |
| 7. | Control | Distilled water | MNSUAM |

a final autoextension step of 72 °C for 5 min. After staining with ethidium bromide (100 μg/ml), the amplified DNA was separated on a 1.0% (w/v) agarose gels (0.5 × TBE buffer) and subsequently observed under an ultraviolet trans-illuminator. The pure product of PCR was sent to MicroGen for sequence analysis.

## Management of *Pa* with chemicals

The assessment of three concentrations (100, 200, and 300 ppm) of each of the six bactericide and copper-based fungicides (Air One, Blue Copper, Cordate, Profiler, Spot Fix and Strider) (Table 1) and control (distilled water) were conducted in the laboratory by using the inhibition zone technique. In order to achieve these concentrations, a sufficient amount of stock solution for all chemicals was prepared by following the protocol provided by *Rehman et al. (2015)*.

## *In-vitro* evaluation of different chemicals

For the management of *P. atrosepticum*, the *in-vitro* inhibition zone technique was followed by different chemicals (given below) with different concentrations (100, 200, 300 μL) under controlled conditions. The purified strains of bacterium were streaked on NA media plates using a sterilized loop. A piece of blotter paper having a 1.5 cm diameter was dipped into chemical solutions (as mentioned above list) at various concentrations (100, 200, 300 μL) and placed with the help of forceps in the center of homogenously streaked bacterial culture. Three replications for each treatment were applied following a completely random design (CRD). In the control treatment, the blotter paper was dipped into the distilled water. Plates were wrapped with the parafilm and incubated at 28 °C and digital vernier caliper was used to measure the corresponding inhibitory zone at 12, 24, and 36 h interval (*Ali et al., 2022*).

## *In-vivo* evaluation of different chemicals

After the evaluation in the lab, these chemicals were also tested on plants located at MNS-University of Agriculture, Multan. 21 pots were prepared that were 12 inches deep and wide. Each pot was filled with a mix of compost and potting soil. Potato seeds were planted in the greenhouse by sowing the seed tubers 4–5 inches deep into the soil of the pots, with the eyes pointing up. After 3 weeks of stem emergence, all plants were watered

properly and covered with polythene bags for 2 h and kept under the sunlight to create high humid conditions and for maximum opening of stomata for inoculation. Inoculation was done in the morning in the greenhouse by using the syringe method. Suspension of the bacteria (about 2 μL) was injected into the stem of the experimental plant (*Francis, Peña & Graham, 2010*). Inoculated potato plants were treated with an injection of six chemicals (about 2 μL) in the stem of the plant at three different concentrations (100, 200, and 300 ppm) by using injection. The plants were monitored every 24 h intervals and data was recorded.

## Statistical analysis

A completely randomized design (CRD) was used to measure the stem length affected and inhibition zone in *in-vivo* conditions. The analysis of variance (ANOVA) was performed at a 5% significance level on the recorded data. Fisher's least significant difference (LSD) test was used in context to ANOVA to compare the mean of all treatments formed lesion length and inhibition zone (*Steel, Torrie & Deekey, 1997*).

# RESULTS

## Survey and isolation

The characteristic symptoms of disease (slimy, wet, black rot lesions appearing on the tuber and spreading from the infected tuber up to the stem) were observed during sample collection. A total of 10 isolates were isolated from collected samples of blackleg of potato. Out of 10 isolates, six isolates viz., PCA-1, 2, 4, 5, and 6 exhibited light yellowish pigmentation, two isolates viz., PCA-7 and 8 exhibited whitish pigmentation, and rest two isolates viz., PCA-9 and 10 exhibited yellowish pigmentation (Fig. 1).

## Biochemical tests

For identification and biochemical characterization, the bacterial isolates were subjected to different biochemical tests. The isolates were determined to be Gram-negative by showing a pinkish color under the microscope. The test bacterium showed a positive reaction to KOH by producing a thread-like string and was determined to be Gram-negative by showing a pinkish color under the microscope in gram staining test.

## Pathogenicity

### Koch's postulates

Characteristic symptoms (slimy, wet, black rot lesions appearing on the tuber and spreading from the infected tuber up to the stem) of this disease were observed on the Potato plants 4 days after they were inoculated (Fig. 2). These symptoms closely resembled those observed in the potato with a black leg in the field during our surveys. The pathogenicity was performed by re-isolating and re-purification from the inoculation potato plants and confirmed Koch's postulates.

## Tuber assay

Following an incubation period of 3–4 days, the tubers were subsequently sliced from a cross-sectional area and assessed for signs of deterioration and infection. The inoculated

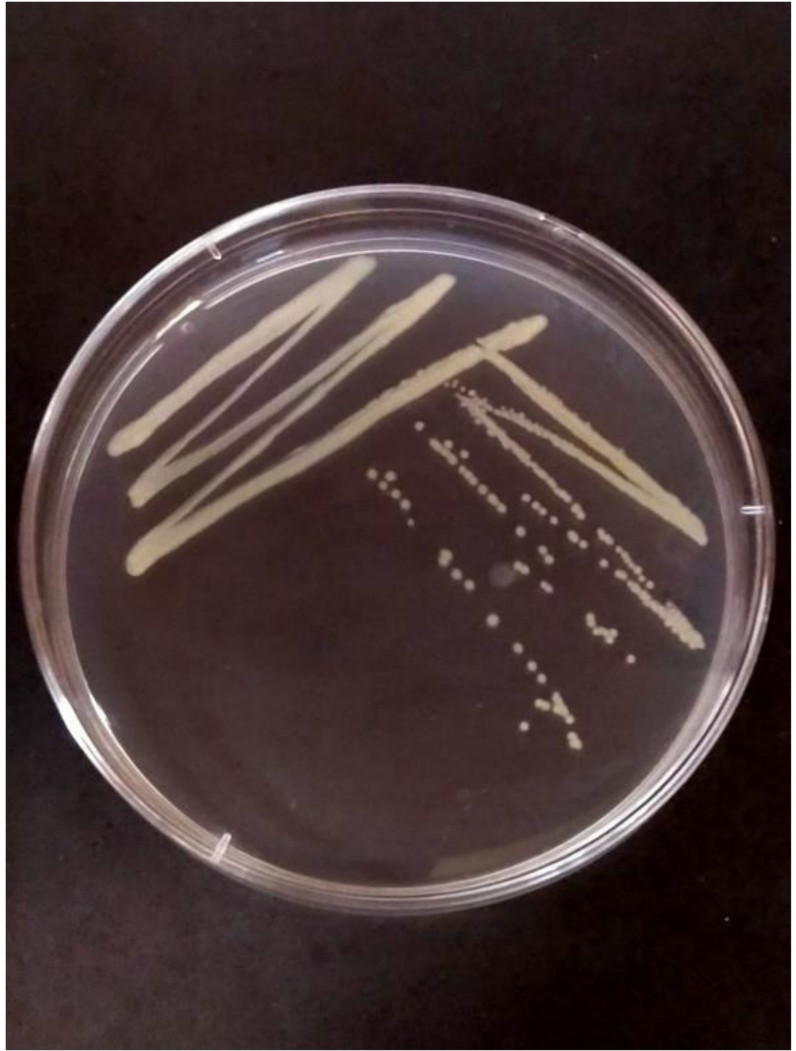

**Figure 1** **PCA-1 isolate purified petri plate showing *Pa* growth.**

tubers had a significant level of tuber maceration and rotting, which is a common sign of a blackleg disease caused by *Pa*. In contrast, there were no signs of deterioration on the tubers in the control group (Fig. 3).

## Molecular characterization

The DNA of the three most aggressive isolates (PCA-5, PCA-7, and PCA-10) was extracted using molecular techniques. The obtained sequences with accession numbers (MK392513.1, MK392518.1 and MK392519.1) were BLAST and confirmed the association between isolated *Pa* with other *Pa* strains. Some closely related sequences were obtained and utilized to construct a phylogenetic tree (Fig. 4).

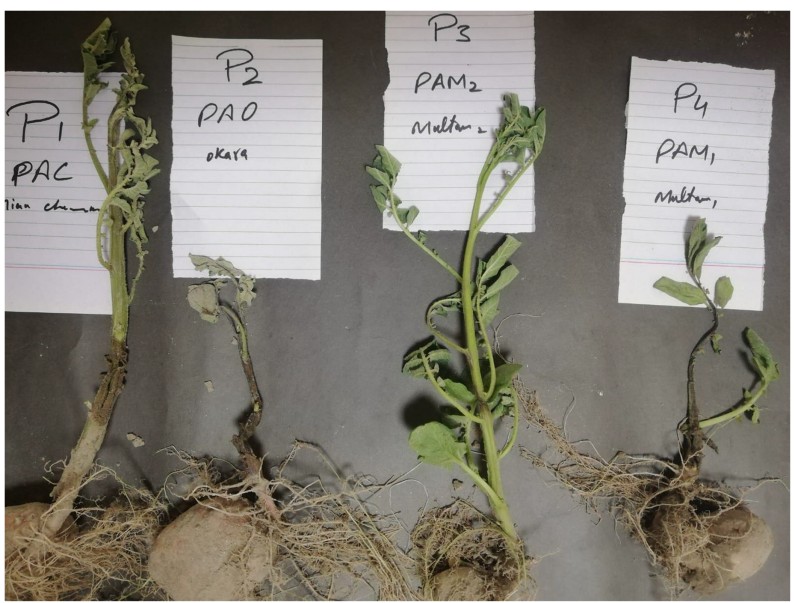

**Figure 2 Observed symptoms of black leg after 4 days of inoculation.**

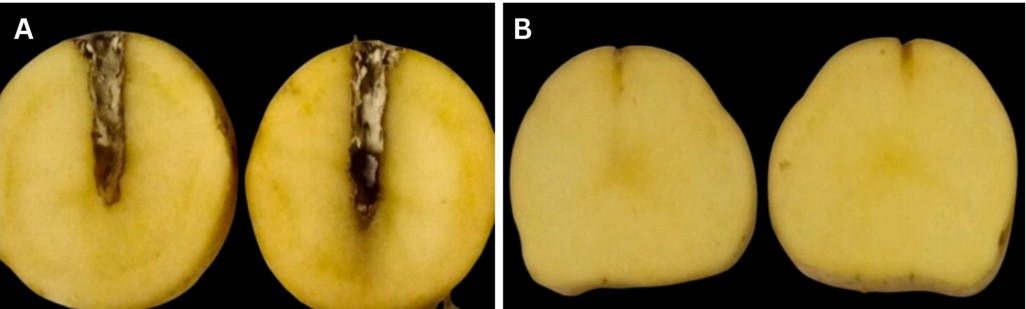

**Figure 3 (A) Tuber assay pathogenicity test shows decaying symptoms, (B) control tuber assay pathogenicity test shows no symptoms.**

## Management of *Pa* with chemicals

A significant difference in efficacy was observed among these chemicals at different doses against the colony growth of *Pa*. The largest inhibition zones were observed with Cordate at 17.139 mm, followed by Air One at 13.778 mm, Profiler at 10.167 mm, Blue Copper at 7.7778 mm, Spot Fix at 7.6689 mm, and Strider at 7.0667 mm as compared to control by these chemicals as shown in Figs. 5 and 6.

The interaction between treatments and concentrations (T × C) showed that the maximum inhibition zone (19.166 mm) was produced by Air One at 200 ppm, (8.166 mm) at 100 ppm and (14 mm) at 300 ppm respectively while Strider exhibited minimum inhibition zones of (6.633 mm), (7.133 mm) and (7.433 mm) at 100, 200 and 300 ppm concentration, respectively 'Fig. 7'.

It is indicated that all the treatments (T), concentrations (C), and their interactions (T × C) exhibited significant results. In the study, Cordate exhibited the lowest disease

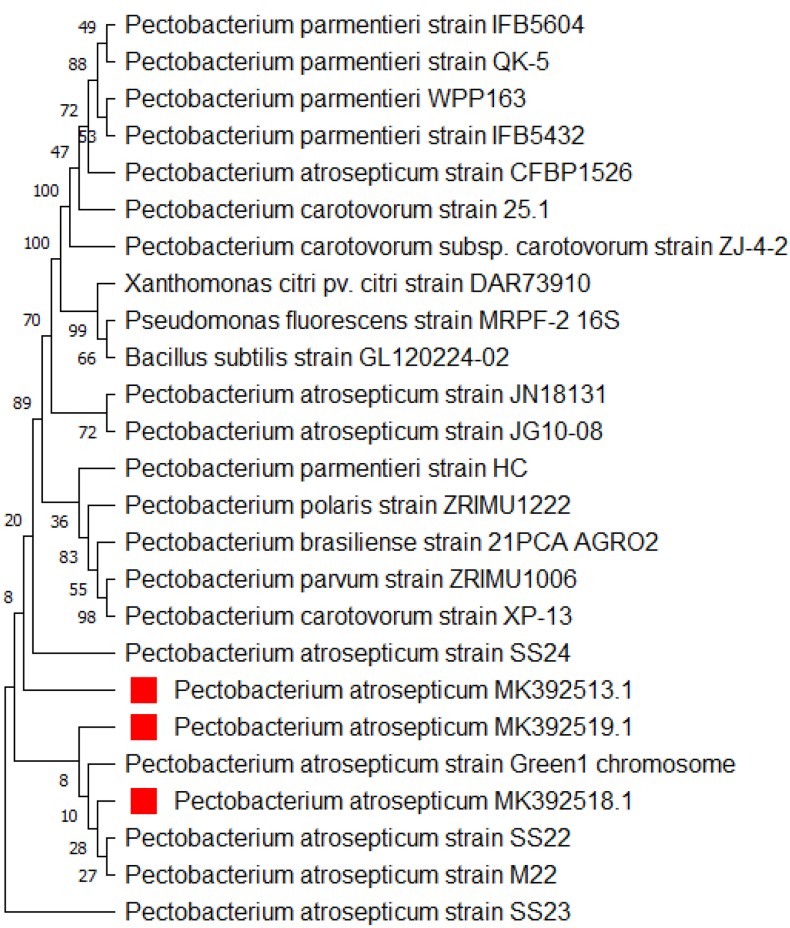

**Figure 4** Dendrogram representing genetic relationships between isolated PCA-5, PCA-7 and PCA-10 with the strains of *Pa*.

incidence at 14.76%, with Blue Copper following at 17.49%, Air One at 16.98%, Spot Fix at 20.67%, Profiler at 21.45%, Strider at 24.99%, and the control group at 43.00% 'Fig. 8'. The interaction between T × C exhibited that minimum disease incidence (11.89%) was produced by Cordate at 300 ppm, (16.73%) at 100 ppm and (15.64%) at 200 ppm respectively while Strider exhibited maximum disease incidence of (29.17%) (23.81%) and (21.95%) at 100, 200 and 300 ppm concentration, respectively 'Fig. 9'.

## DISCUSSION

Potatoes have a high market value because they include an abundance of essential nutrients such as carbohydrates, minerals, vitamins, and proteins. Losses in yield have a dual effect on society, resulting in both food insecurity and economic turmoil (*Geddes et al., 1989*; *Ali et al., 2022*). Potato growers face a significant threat from phytopathogens like *Pectobacterium atrosepticum* (the causative agent of blackleg disease). When a disease invades a plant, it causes the stems and tubers to decay softly. When a plant becomes sick from the ground up, its upper portions wither and die. Blackleg is so named because it causes slowed growth and a blackening of the lower stem (*Ma et al., 2018*). Disease

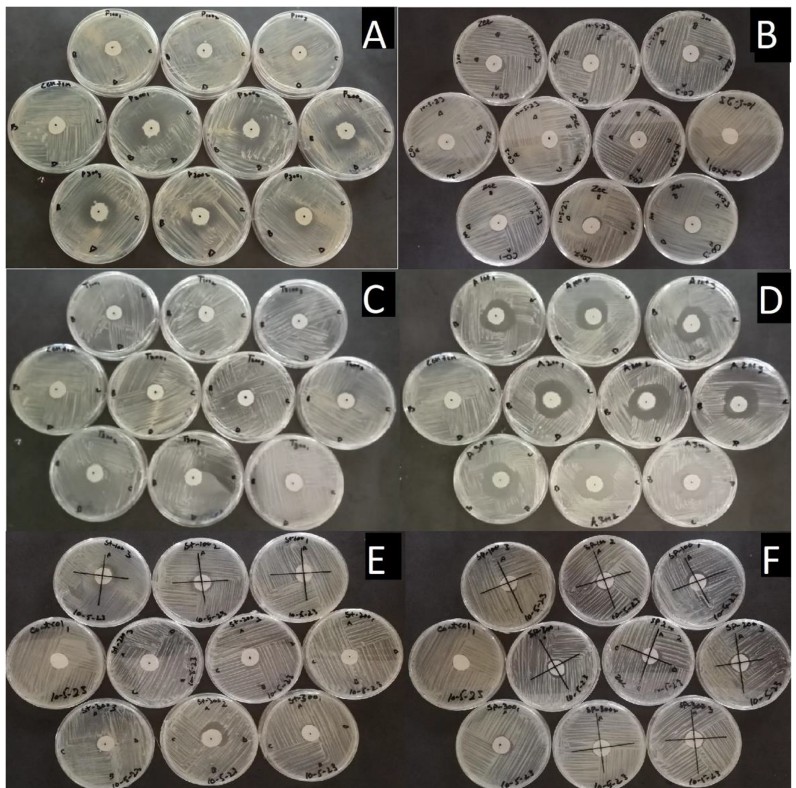

**Figure 5** Pictorial view of inhibition zones formed by the chemicals (A) Air One, (B) Blue Copper, (C) Profiler, (D) Cordate, (E) Strider, (F) Spot Fix with control *in-vitro* application for the management of *Pa*.               

pyramid components including susceptible host, virulent pathogen, and favorable temperature are compulsory to develop a disease. Even if one component is absent, there is no disease. The development of disease depends on epidemiological factors and host-pathogen relations (*Czajkowski, Grabe & van der Wolf, 2009*).

The nutrient agar media was used to evaluate the morphological characteristics of ten distinct test isolates. Colony color and shape were among the characteristics that were observed. Six of the ten isolates PCA-1, 2, 3, 4, 5, and 6 showed a light yellowish pigmentation, whereas PCA-7 and 8 showed a whitish pigmentation, and the last two isolates PCA-9 and 10 showed yellowish pigmentation. Each tested isolate had a regular colony morphology growth on NA media. These results are consistent with earlier studies carried out by *Cinisli et al. (2019)*.

In the KOH experiment, viscous strands of material revealed the bacterium is Gram-negative. This result was in line with that of previous research conducted by *Ali et al. (2022)*, which additionally confirmed the Gram-negative classification of the *Pa* bacterium through this method. In gram staining, the results revealed that the test bacterium had lost its ability to maintain the violet color of the primary stain (Crystal violet), but that cells had taken on a pink type as a result of counterstaining with the stain safranin. As a result, the test bacterium had a gram-negative, straight rod shape which is a typical feature of bacteria that cause blackleg disease. Similar results were observed during the Gram staining test

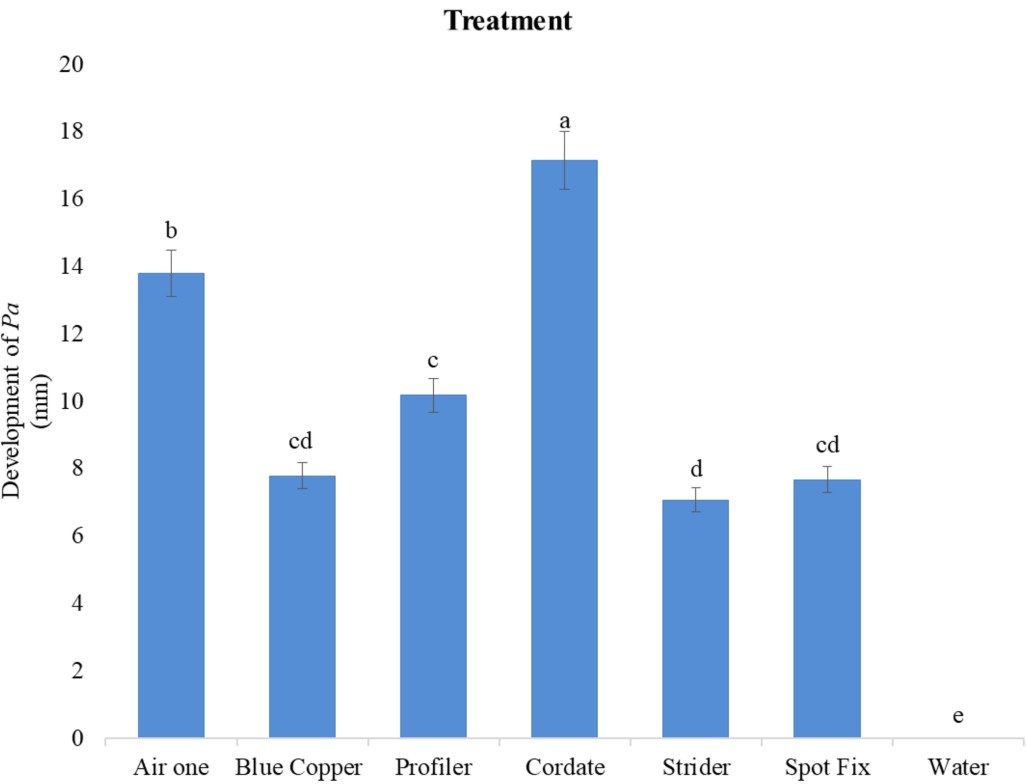

**Figure 6** *In-vitro* evaluation of various chemicals on the development of *Pa*.

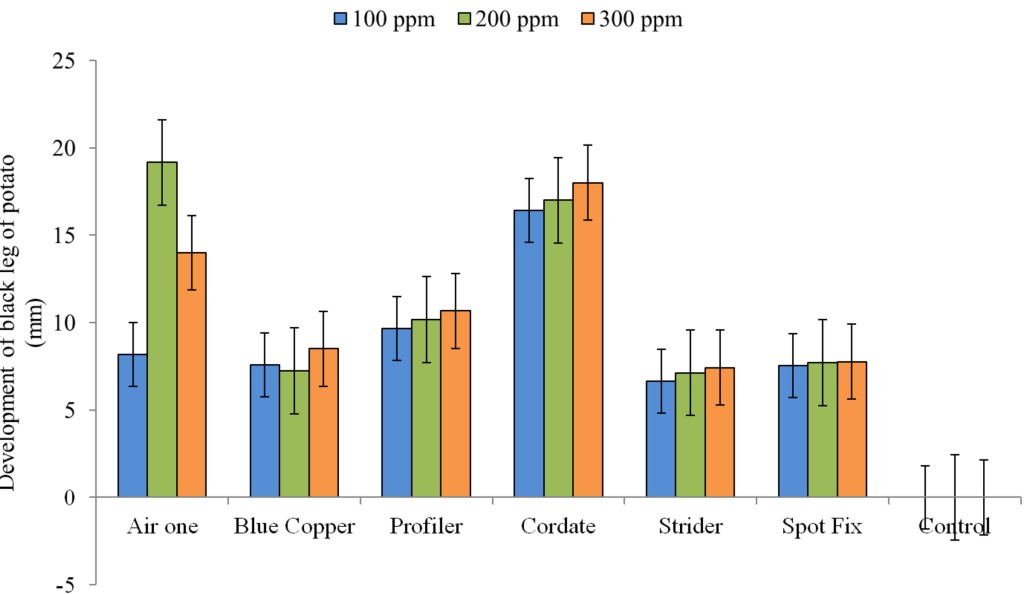

**Figure 7** Impact of interaction b/w treatments and concentrations (T × C) on the development of black leg of potato.

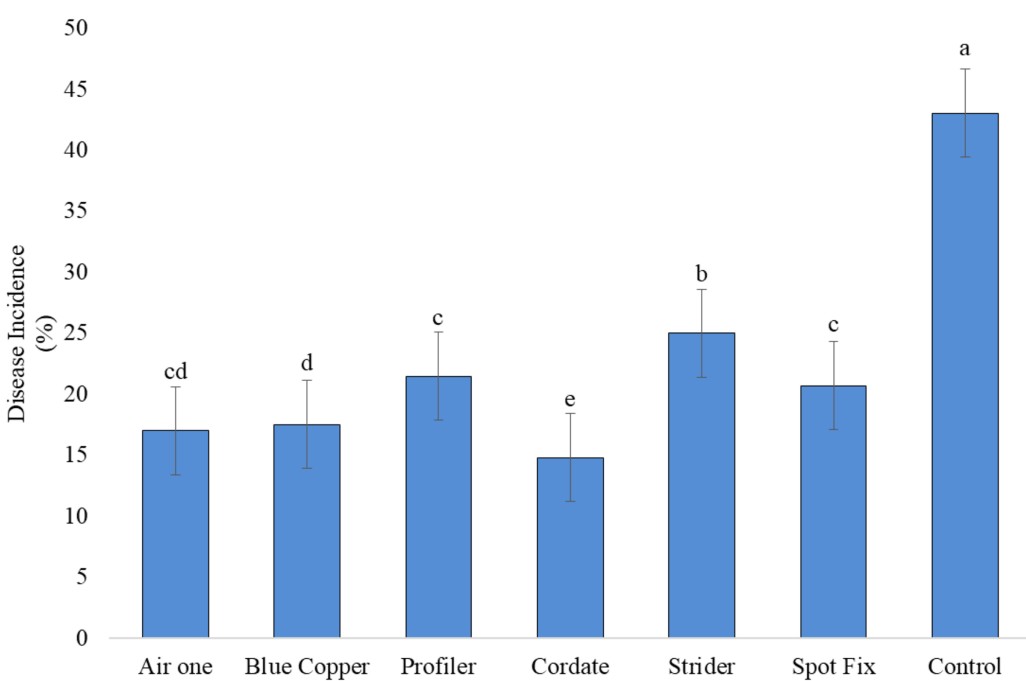

**Figure 8** Evaluation of different antibiotic against *Pa* under greenhouse conditions.

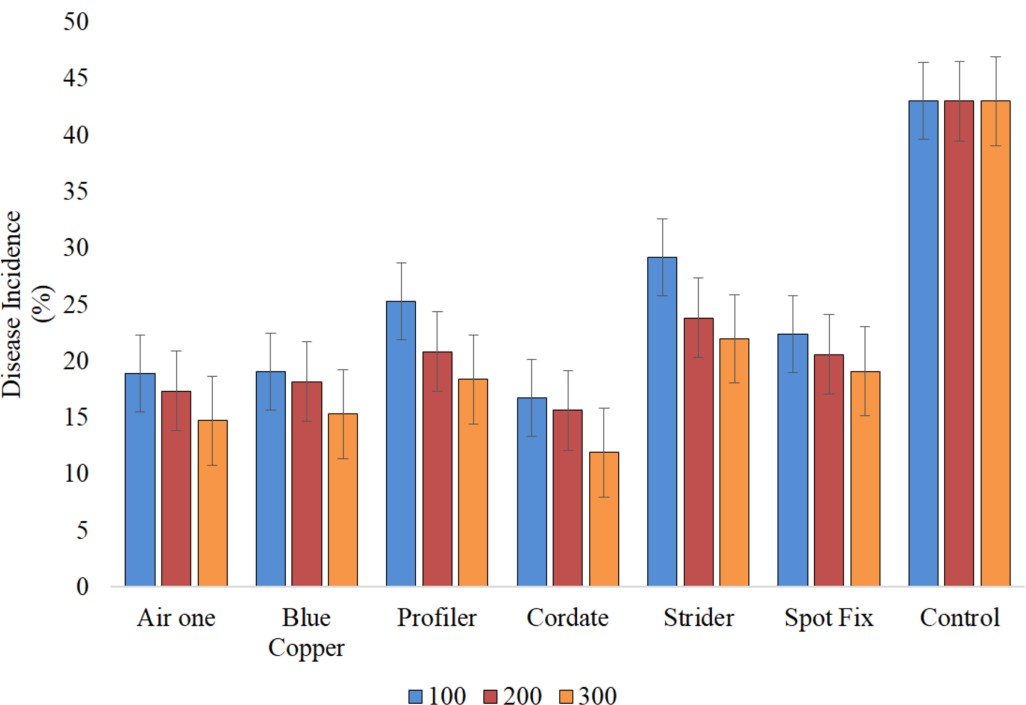

**Figure 9** Evaluation of interaction b/w treatments and concentrations against *Pa* under greenhouse conditions.

performed by *Ali et al. (2022)* who also observed the pinkish color of *Pa* bacterium under the microscope and confirmed that *Pa* was a gram-negative and casual organism of black leg disease.

By Koch's postulates, the pin-prick method was used to inoculate the bacterial suspension on healthy potato plant stem for a pathogenicity test. Following the suspension inoculation, symptoms began to manifest after 4 days, indicating the development of pathogenic effects. These observed symptoms were identical to the symptoms observed in the field during the survey. Tuber assay testing was performed on isolate PAC-1. After 3–4 days, *P. atrosepticum* infected tubers showed severe rotting and maceration. The control tubers showed no deterioration. The pathogenicity findings were in line with the results from previous research conducted by *Ma et al. (2018)*, who also noted the pathogenicity of *Pa* by inoculating on potato plants.

The DNA of the most aggressive isolates was extracted and analyzed using PCR targeting the 16s rDNA gene. Gel electrophoresis with fluorescent light was used to observe a DNA fragment of 1,500 base pairs. After identification, the DNA was commercially sequenced. To perform a phylogenetic analysis of the *Pa* genome sequence, the BLAST tool on NCBI was employed. The BLAST results confirmed the correspondence between the genome sequence and *Pa*, confirming that this pathogen is the causal agent of the blackleg of the potato. Additionally, closely related sequences were downloaded and used to study the relationship between related pathogens. An evolutionary tree was then developed using these sequences. The relationship between this evolutionary tree and other blackleg of potato diseases was revealed. This finding similar to the study conducted by *Ali et al. (2022)*, who confirmed after sequence that all their isolates of *Pa* strains were closely related to the *Pa* strains found in the gene sequence data of *Pa* strains from the NCBI blast in the phylogenetic tree analysis.

After identification of the pathogen, several actions can be taken to control the disease, for instance, the use of resistant potato varieties (*Chiesa et al., 2013*; *Lu et al., 2022*), the design of effective control measures based on the genetic diversity, population characteristics, and virulence of *Pa* (*Izadiyan, Taghavi & Farahbakhsh, 2018*; *Timilsina et al., 2020*; *Webster, Bogema & Chapman, 2020*), and the determination of the gene content differences within the genetic groups (*Richard et al., 2017*). Through the implementation of these measures blackleg can be prevented from spreading. The research described herein focuses on the six chemicals that are effective in fighting the disease. Six complex chemical compounds were utilized to performed the experiment in three different concentrations and both *in-vivo* and *in-vitro* situations. Cordate proved to be the best chemical among all the tested ones both *in vitro* and *in vivo* in preventing the development of *Pa*. The present study was in line with research described by *Czajkowski et al. (2011)*, who studied how various chemicals affected the blackleg of potato. They concluded that Cordate had the best results in reducing Pectobacterium. By attaching to the bacterial chromosomes, Cordate inhibits bacterial reproduction and functions as a bacteriostatic drug. In addition, it has been documented that copper-based pesticides exhibit a certain level of phytotoxicity towards plants. Furthermore, the excessive use of

these chemicals has been linked to the development of resistance in several bacterial species (*Lalancette & McFarland, 2007*; *Águila-Clares et al., 2018*).

## CONCLUSIONS

This study was designed to determine a morphogenic characterization of *Pa* and the management of it through new chemicals. On the basis of morpho-molecular characterization and pathogenicity assay, it was confirmed that the causal agent of potato blackleg disease was *Pectobacterium atrosepticum*. Six chemicals were tested *in vitro* and *in vivo* to assess their efficacy against *Pa*. Cordate at 300 ppm concentration showed the maximum effectiveness in preventing the development of *Pa* both *in-vitro* and *in-vivo* conditions.

### Funding

The authors received funding from the Researchers supporting project number (RSP2024R185), King Saud University Riyadh, Saudi Arabia. The funders had no role in study design, data collection and analysis, decision to publish, or preparation of the manuscript.

### Grant Disclosures

The following grant information was disclosed by the authors:
King Saud University Riyadh, Saudi Arabia: RSP2024R185.

### Competing Interests

The authors declare that they have no competing interests.

### Author Contributions

- Akhtar Hameed conceived and designed the experiments, prepared figures and/or tables, authored or reviewed drafts of the article, and approved the final draft.
- Muhammad Zeeshan performed the experiments, prepared figures and/or tables, and approved the final draft.
- Rana Binyamin conceived and designed the experiments, performed the experiments, prepared figures and/or tables, authored or reviewed drafts of the article, and approved the final draft.
- Muhammad Waqar Alam conceived and designed the experiments, analyzed the data, prepared figures and/or tables, and approved the final draft.
- Subhan Ali performed the experiments, analyzed the data, prepared figures and/or tables, authored or reviewed drafts of the article, and approved the final draft.
- Muhammad Saqlain Zaheer conceived and designed the experiments, performed the experiments, authored or reviewed drafts of the article, did data analysis, and approved the final draft.
- Habib Ali performed the experiments, analyzed the data, prepared figures and/or tables, authored or reviewed drafts of the article, and approved the final draft.

- Muhammad Waheed Riaz performed the experiments, analyzed the data, prepared figures and/or tables, authored or reviewed drafts of the article, and approved the final draft.
- Hafiz Haider Ali performed the experiments, analyzed the data, authored or reviewed drafts of the article, and approved the final draft.
- Mohamed Soliman Elshikh conceived and designed the experiments, analyzed the data, prepared figures and/or tables, and approved the final draft.
- Khaloud Mohammed Alarjani conceived and designed the experiments, performed the experiments, analyzed the data, authored or reviewed drafts of the article, and approved the final draft.

## Data Availability

The raw data is available in the Supplemental Files.

## Supplemental Information

Supplemental information for this article can be found online at http://dx.doi.org/10.7717/peerj.17518#supplemental-information.

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
