# Peer review of "Molecular characterization of Pectobacterium atrosepticum infecting potato and its management through chemicals"

_PeerJ, doi:10.7717/peerj.17518_

## Round 0.1 · original submission · Major Revisions

The whole manuscript should be checked thoroughly by an English expert, the manuscript lacks clarity and novelty, authors need to address all the reviewers' queries. The manuscript in its current form cannot be accepted for publication. The contribution of the research should be given.

**Language Note:** The review process has identified that the English language must be improved. PeerJ can provide language editing services - please contact us at [email protected] for pricing (be sure to provide your manuscript number and title). Alternatively, you should make your own arrangements to improve the language quality and provide details in your response letter. – PeerJ Staff

·

Basic reporting

Although the manuscript demonstrates proficiency in professional language and structure, there is an opportunity for improvement in enhancing clarity, linking references to the study, strengthening the connection of figures and tables to hypotheses, and explicitly providing information about the availability of raw data.

Experimental design

The approach taken in the experimental design is thorough. Still, specific aspects such as criteria for sample selection, biochemical tests, greenhouse conditions, and the rationale for chosen methods require further clarification. Providing these details will augment the strength and transparency of the experimental design.

Validity of the findings

The study's validity is well-supported by a sound methodology and statistical analysis, and transparency in data sharing. However, specifying data availability and ensuring that conclusions are closely tied to the research question and results will enhance the manuscript's overall coherence.

Additional comments

Disease severity has not been calculated or reported. Disease severity is an important parameter in plant pathology studies as it helps in understanding the impact of the disease on the host plants. It would be helpful to add a column on disease severity next to the disease incidence in the table of data analysis provided in supplementary materials.

Reviewer 2 ·

Basic reporting

1. The introduction could benefit from better organization and structure. It currently presents information in a somewhat fragmented manner, making it challenging for readers to follow the flow of information.
2. Some citations need to be updated; more recent sources are recommended.
3. The significance of potato farming and its potential impact on food security and poverty reduction was addressed, but it could be expanded on this aspect. E.g., why it's crucial to manage blackleg disease and how it relates to these broader goals.
4. Clearly introduce Pectobacterium atrocepticum, responsible for causing Blackleg disease, as the focal point of the study.
5.Provide background information on the characteristics and impact of Pectobacterium

Experimental design

Survey:
Specify the survey duration to provide context for when the field data was collected.
Consider providing a brief overview of the number of fields surveyed and the criteria for selecting them.
Pathogenicity Tests:
Specify the number of replicates used in Koch's postulates and the Tuber assay test.
Provide details on the greenhouse conditions (e.g., temperature, light) during the pathogenicity tests.
DNA extraction and PCR:
Specify the specific primers used and their source.
Management of Pa with Chemicals:
Provide information on the number of bactericides and copper-based fungicides tested.
Clarify the rationale behind choosing the concentrations (100 ppm, 200 ppm, and 300 ppm).

Validity of the findings

Pathogenicity Tests:
The pathogenicity tests, including Koch's postulates, are crucial for establishing the causal relationship between the isolated bacterium and the observed symptoms. Including a detailed timeline for symptom development after inoculation and comparing symptoms observed in the field surveys would strengthen the argument.
Molecular Characterization:
It would be beneficial to provide more specifics about the universal primers used and the size of the PCR product obtained.
Are the DNA sequences submitted to any databank? If so, including an accession number is recommended.
Some figures (7,8,9) mentioned in the text are missing,

Additional comments

the

Reviewer 3 ·

Basic reporting

English language improvements as an example:
The abstract first sentence English needs to be more concise
“Potato is an important food and cash crop that production losses” (maybe change to whose production or cultivation)
“After biochemical and pathogenicity tests isolated bacteria were gram-negative” (maybe change to “were observed to be “)
Kindly amend such changes

Experimental design

Literature review is sufficiently covered for a basic study.
But the work is deemed basic in my opinion as novelty in the study is missing.
The work seems like a survey of existing chemical pesticides.

Validity of the findings

How is the finding different from the USEPA findings of the usage / detection of kasugamycin
https://www3.epa.gov/pesticides/chem_search/reg_actions/registration/fs_PC-230001_01-Sep-05.pdf
“Kasumin® 2L, a liquid formulation comprised of 2% kasugamycin (by weight) as the active
ingredient (ai), for use on rice, potato, pepper, and tomato in Mexico.”

---

## Round 0.2 · accepted · Accept

The authors have addressed all of the reviewers' comments, the manuscript is ready for publication.

·

Basic reporting

No further comments.

Experimental design

No further comments

Validity of the findings

No further comments

Additional comments

No further comments

Reviewer 2 ·

Basic reporting

The manuscript has significant improvements from the previous submission. Proficiency and clarity of the literature have been enhanced. The methodology and analysis method have been made clearer. The results are now concise and focused, providing a clear interpretation of the findings. Additionally, the authors have addressed previous concerns regarding sample size, control measures, and potential limitations, thereby strengthening the validity of the study's results. The manuscript now presents a cohesive narrative, guiding the reader through each step of the research process with clarity and precision.

Experimental design

Suggestions provided during the review process have been addressed in the methodology section. The isolation and analysis procedures have been significantly clarified, rendering them more accessible and comprehensible to readers.

Validity of the findings

The study makes a robust contribution to the field, addressing important questions regarding blackleg disease in potato plants. The analytical methods employed demonstrate soundness and reliability, ensuring the validity of the research outcomes. While there may be a few minor details that still need to be addressed,, the manuscript in its current state is deemed acceptable for publication.

Additional comments

I recommend proceeding with the publication of the manuscript, confident that it will contribute meaningfully to the scientific discourse in the field of plant pathology.